# Time-Dependent Changes in Muscle IGF1-IGFBP5-PAPP System after Sciatic Denervation

**DOI:** 10.3390/ijms241814112

**Published:** 2023-09-14

**Authors:** Ana Isabel Martín, Álvaro Moreno-Rupérez, Elena Nebot, Miriam Granado, Daniel Jaque, M. Paz Nieto-Bona, Asunción López-Calderón, Teresa Priego

**Affiliations:** 1Departamento de Fisiología, Facultad de Medicina, Universidad Complutense de Madrid, Plaza de Ramón y Cajal sn, 28040 Madrid, Spain; anabelmartin@med.ucm.es (A.I.M.); alvmor02@ucm.es (Á.M.-R.); elenaneb@ucm.es (E.N.); alc@med.ucm.es (A.L.-C.); 2Departamento de Fisiología, Facultad de Medicina, Universidad Autónoma de Madrid, Calle Arzobispo Morcillo 2, 28029 Madrid, Spain; miriam.granado@uam.es; 3Nanomaterials for Bioimaging Group (NanoBIG), Departamento de Física de Materiales, Facultad de Ciencias, Universidad Autónoma de Madrid, Avenida Francisco Tomas y Valiente, 28049 Madrid, Spain; daniel.jaque@uam.es; 4Departamento de Ciencias Básicas de la Salud, Facultad CC Salud, Universidad Rey Juan Carlos, Avenida de Atenas sn, 20922 Madrid, Spain; paz.nieto@urjc.es; 5Departamento de Fisiología, Facultad de Enfermería, Fisioterapia y Podología, Universidad Complutense de Madrid, Plaza de Ramón y Cajal sn, 28040 Madrid, Spain

**Keywords:** IGF-1, IGFBP-5, skeletal muscle atrophy, pappalysins

## Abstract

Denervation-induced muscle atrophy is a frequent cause of skeletal muscle diseases. However, the role of the most important muscle growth factor, insulin-like growth factor (IGF-1), in this process is poorly understood. IGF-1 activity is controlled by six IGF-1 binding proteins (IGFBPs). In skeletal muscle, IGFBP-5 seems to have an important role in atrophic processes. Furthermore, pappalysins (PAPP-A) modulate muscle growth by increasing IGF-1 bioavailability through IGFBP cleavage. We aimed to study the time-dependent changes in the IGF1-IGFBP5-PAPP system and its regulators in gastrocnemius muscle after sciatic denervation. Gastrocnemius atrophy and overexpression of IGF-1 was observed from day 3 post-denervation. The proteolytic factors measured were elevated from day 1 post-denervation onwards. Expression of both IGFBP-5 and pappalysins were increased on days 1 and 3. Subsequently, on days 7 to 14 pappalysins returned to control levels while IGFBP-5 remained elevated. The ratio IGFBP-5/PAPP-A was correlated with the main proteolytic markers. All data suggest that the initial increase of pappalysins could facilitate the IGF-1 action on muscle growth, whereas their subsequent decrease could lead to further muscle wasting.

## 1. Introduction

Skeletal muscle is involved in movement and strength, but it also has a wide range of endocrine and metabolic actions. In this way, many clinical conditions such as malnutrition, aging, neurodegenerative diseases, sepsis, obesity, or diabetes, among others, occur with muscle impairment [1,2]. Sciatic denervation is a widely used model of muscle atrophy induced by peripheral nerve damage. Although substantial advances have been made in muscle atrophy induced by denervation over the last decade [3], even using new omics techniques [4,5], few studies have examined denervation-induced changes in the major factors involved in muscle protein synthesis and fiber growth.

Insulin-like growth factor 1 (IGF-1) is one of the most important growth factors in skeletal muscle. Not only is it essential for muscle development during growth, but also for regeneration and adaptation in mature muscle fibers [6]. Although circulating IGF-1 is mainly synthesized by the liver, other tissues such as skeletal muscle can produce it. In fact, it has been reported that local production appears to be more important for muscle regeneration than hepatic IGF-1 [7]. Hambretch et al. (2005) [8] observed that exercise increases muscle IGF-1, whereas they did not find changes in other tissues. Moreover, it has been demonstrated that muscle IGF-1 deletion induces muscle weakness and atrophy [9].

The IGF-1 half-life (10–12 min) can be prolonged to 12–16 h, by the binding of this factor to a family of binding proteins called IGF-binding proteins (IGFBPs). In addition, these proteins also regulate IGF-1 bioavailability [10,11]. In this respect, it has been established that the IGFBP’s function is essential for the IGF-1 system signaling pathway [12]. Among the six members of the IGFBPs family, IGFBP-5 is the most abundant in skeletal muscle [13]. IGFBP-5 exhibits a diverse range of biological effects. It can influence IGF-1 activity through several mechanisms (reviewed in [14]). First, it can extend the half-life of IGF-1 in the bloodstream, allowing for a longer duration of its actions. In addition, it can compete with the IGF-1 receptor for ligand binding, thereby inhibiting IGF-1 signaling. However, it can also do the opposite; it can enhance IGF-1 signaling by aiding in the delivery of IGF-1 to the IGF-1 receptor. It can also concentrate IGF-1 in specific cells and tissues, facilitating localized effects. Furthermore, IGFBP-5 displays IGF-independent activities. Whether IGFBP-5 acts in one way or another will depend on the type of cell and the physiological or pathological context in which it operates [15]. This highlights the complexity of IGFBP-5’s functions and the importance of considering the specific conditions under which it is acting. In relation to muscle atrophy, several studies have pointed out the involvement of IGFBP-5 in muscle wasting. For instance, in muscle atrophy induced by disuse, such as immobilization or hindlimb suspension, increased expression of muscle IGFBP-5 has been observed [16]. Additionally, elevated IGFBP-5 levels have been reported in muscle wasting induced by chronic inflammatory diseases, such as arthritis [17]. We have also described an increase in IGFBP-5 in age-related sarcopenia [18]. There are a few studies on muscle IGFBP mRNA expression in denervation-induced muscle atrophy. IGFBP-5 upregulation has been reported 2 days after denervation [19]. These authors suggested that IGFBP-5 may play a key role in denervation-induced muscle atrophy. Nevertheless, the time-dependent response of IGFBP-5 after denervation needs further research to elucidate its possible role in muscle atrophy.

There is another family of proteins involved in the IGF-1 system. Lawrence et al. [20] identified pregnancy-associated plasma protein-A (PAPP-A) or pappalysin as a IGFBPs protease. Later, interactions between IGFBP-5/PAPP-A [21] and IGFBP-5/PAPP-A2 were described [22]. In muscle, several pieces of evidence indicate that PAPP-A increases local IGF-1 bioavailability. PAPP-A not only has an impact on postnatal growth [23,24], but it also plays a key role in the myogenic program and in prenatal development [25,26]. In a long-lived PAPP-A KO mouse, increased mouse lifespan and enhanced muscle function were reported, suggesting that PAPP-A seems to have an important role in skeletal muscle physiology [27]. Pappalysins are inhibited by other proteins, the stanniocalcins. Stanniocalcins (STC-1 and 2), which are widely expressed in many tissues [28], have the ability to bind with PAPP-A and to inhibit its proteolytic activity. Their role as potent inhibitors of pappalysins has been demonstrated in cell cultures and in transgenic animals [29]. Even though STC-2 has been described as a negative modulator of human growth [30], its specific function in muscle physiology has never been investigated.

Knowing that IGF-1 could be a possible therapeutic target for the treatment of muscle atrophy in muscular diseases related to the nervous system [31], we aimed to better understand the time-dependent changes in the IGF-1 system after denervation, including the analysis of IGFBP-5 and its regulators, PAPP and STC. These results may have important implications for the treatment of nerve-related muscular atrophy.

## 2. Results

### 2.1. Denervation Effect on Gastrocnemius Muscle Atrophy and Myofiber Isoform Expression

In order to study the effects of denervation on muscle atrophy, the sciatic nerve of the right hind paw was sectioned, and the gastrocnemius muscle was analyzed and compared with that of the left hind paw at different times. As previously described [32], gastrocnemius weight and the mean fiber cross-sectional area were decreased by denervation, being significantly lower than their respective contralateral hind paw 7 days after denervation (Figure 1A,B, *p* < 0.01 with Student’s *t* test). On day 14 after denervation, the gastrocnemius weight was approximately 46% of the contralateral hind paw, whereas the fiber area was very low, decreasing to 26% of the control values.

As shown in Figure 1C,D, the myosin heavy chain I (MyHC-I) mRNA decreased 7 days after denervation onwards (*p* < 0.05 with Student’s *t* test), but no significant modification was observed in the MyHC IIb mRNA.

### 2.2. Denervation Effect on the IGF1-IGFBP5-PAPP System

The IGF-1 mRNA levels increased in the gastrocnemius three days after denervation (Figure 2A, *p* < 0.001 with Student’s *t* test). This declined over time, although, at both 7 (*p* < 0.05) and 14 days (*p* < 0.001) post-denervation, levels of this factor remained significantly elevated in comparison to those of their respective contralateral muscles. The expression of the IGF-1 receptor (IGF-1R) followed a similar pattern to that of IGF-1. It was characterized by an important increase at day 3 post-denervation (*p* < 0.05 with Student’s *t* test, Figure 2B). However, in this case, the expression of this receptor did not significantly decrease during the following days.

IGFBP-5 levels were significantly elevated from day 1 to 14 days post-denervation (*p* < 0.001, Figure 2C). 4E-BP1 (eukaryotic initiation factor 4E-binding protein 1), also known as EIF4BP1, acts as a negative regulator of protein synthesis. Interestingly, the expression levels of this factor increased significantly on day 3 after denervation and decreased on the following days (*p* < 0.05 with one-way ANOVA, Figure 2D).

Figure 2E,F show the expression of the main proteolytic enzymes of IGFBP-5, PAPP-A and PAPP-A2. PAPP-A mRNA levels increased the first day after denervation (*p* < 0.05 with Student’s *t* test, Figure 2E) and declined afterwards (days 3, 7 and 14 post-denervation), reaching similar levels to those of sham muscles. A similar pattern was observed in PAPP-A2 mRNA, although in this case the increase at day one was more pronounced (almost 10-fold) and remained increased on day 3 post denervation (*p* < 0.05 with Student’s *t* test, Figure 2F).

The expression of the PAPP inhibitors STC-1 and 2 is shown in Figure 2G,H, respectively. Neither of the two proteins’ mRNA levels were modified in the gastrocnemius muscle after denervation.

### 2.3. Denervation Effect on Atrophy Mediators

HDAC-4 (histone deacetylase 4) and myogenin are important regulators of muscle proteolysis, since they increase the expression of atrogenes (atrogin-1 and MuRF-1). The mRNA levels of these mediators are shown in Figure 3A,B, and their proteins in Figure 4C,D. HDAC-4 mRNA increased sharply from day 1 post denervation (nearly 10-fold increase, *p* < 0.001 with Student’s *t* test, Figure 3A) and they continued to increase on subsequent days until reaching levels of 20-fold in comparison with the control muscles. Protein levels of HDAC-4 increased on day 3 after denervation and reached their maximum values on day 14 (*p* < 0.05 with one-way ANOVA, Figure 4C).

Similarly, myogenin, atrogin-1 and MuRF-1 (muscle RING-finger protein-1) mRNA increased 1 day after denervation (Figure 3B–D; *p* < 0.05 with Student’s *t* test), but those increases were not as marked (around 2 to 4 fold increase) as that of HDAC-4 mRNA. However, on day 3 post-denervation, myogenin mRNA increased dramatically (up to 30-fold) and these high myogenin levels were also observed on days 7 and 14 post-denervation. The myogenin protein levels increased significantly from day 3 post-denervation onwards (Figure 4D).

The ratio of IGFBP-5/PAPP-A may be used as a reflection of active IGFBP-5. This ratio increased on day 3 after denervation, concomitantly with the upregulation of atrophy markers such as myogenin, and the increased levels were maintained the following days (*p* < 0.05 with one-way ANOVA, Figure 4A). Total protein concentration decreased sequentially after denervation, with the decrease being significant from day 7 (*p* < 0.05 with one-way ANOVA, Figure 4B) onwards. Similarly, on day 7 post denervation, we found significant increases in protein degradation markers such as cleaved alpha II spectrin (marker of calpain activity) (Figure 4E) and LC3b-II (light chain 3b II) (marker of autophagy) (Figure 4F), and these increases were maintained on day 14.

### 2.4. IGFBP-5/PAPP-A Correlations with Atrophic Mediators

Pooling the data from the IGFBP-5/PAPP-A ratio within each time-point of the right (denervated) paw, a simple linear correlation analysis revealed a significant positive correlation between this ratio and atrophic factors such as myogenin (both mRNA and protein levels, r > 0.5, *p* < 0.01, Figure 5A,B), MuRF-1 expression levels (r = 0.499, *p* < 0.01, Figure 5C) and LC3b-II protein levels (r = 0.469, *p* < 0.05, Figure 5D).

## 3. Discussion

IGF-1 plays a central role in skeletal muscle development, maintenance, and regeneration [33]. Alteration of the local IGF-1 system in the skeletal muscle seems to be a common pathological finding in atrophic muscle in different neuromuscular disorders [34]. Our findings indicate that both IGF-1 and its receptor mRNA increase three days after denervation. This increase has been previously reported in both types of muscle, slow (soleus) and fast-twitch (extensor digitorum longus) [35]. In muscles paralyzed by botulinum toxin injection, an increase in IGF-1 mRNA was reported 12 h after exposure to the toxin [36]. The increase in IGF-1 and its receptor may be the consequence of a compensatory mechanism in order to promote anabolic pathways for protecting from muscle atrophy [37,38,39]. The higher percentage of atrophy observed after spinal cord injury (SCI) compared to denervation has been related to the lower increase in IGF-1 expression and its receptor following SCI versus denervation [35]. In fact, the specific over-expression of IGF-1 was able to decrease the rate of muscle atrophy induced by denervation [31]. Similarly, muscle transfection with PKB/Akt (protein kinase B), the main protein involved in the IGF-1 signaling pathway, had a protective effect against atrophy, and it seems to be due to the increased protein synthesis [40]. Likewise, an increase in protein synthesis, as soon as one day after denervation, has been described in denervated diaphragm, which was related with an activation of the IGF-1 system [41].

After muscle denervation, protein synthesis and fiber growth as well as protein de-gradation and atrophy are initiated simultaneously. In fact, the IGF-1 dependent pathway, PKB/Akt-mTORC1 (mammalian target of rapamycin complex 1), is necessary for the initiation of muscle atrophic mechanisms [42]. Activation of the PKB/Akt upon denervation upregulates the HDAC-4-myogenin axis, the main agent involved in denervation-induced atrophy [43]. Our results show that all these factors are involved in protein degradation pathways: HDAC-4, myogenin and the atrogenes (atrogin-1 and MuRF1) increased early, 1–3 days after denervation. In the same way, 4E-BP1 gene expression increased 3 days after denervation. This binding protein is a key downstream effector of mTOR complex that represses cap-dependent mRNA translation initiation by sequestering the translation initiation factor eIF4E [44]. The activation of this protein induces autophagy, and it has been related to muscle atrophy [45]. Therefore, the balance between protein synthesis and degradation seems to lean towards protein degradation around day 7 after denervation, since the fiber area and muscle weight were lower on these days. Concomitantly, atrophic factors such as LC3b-II (marker of autophagy) and those of cleaved alpha II spectrin (marker of calpain activity) also increased on days 7 and 14 after denervation. Additionally, MyHC-I levels, a marker of slow-type fibers, decreased from day 7 after denervation onwards, whereas MyHC-IIb was not modified. This fiber switching from slow to fast fiber is a common process observed in denervated muscle [46]. We could state that, although protein synthesis pathways are activated the first days after denervation (1 to 3 days), later, from day 3 onwards, proteolytic pathways predominate, such as the ubi-quitin-proteasome, autophagy and calpains. The main atrophic factors end up having more weight in the process.

IGFBP-5 has a key role in muscle physiology, regulating both proteolysis and myogenic differentiation [14]. Upregulation of IGFBP-5 has been related to increased proteo-lysis in muscle wasting induced by experimental arthritis [47], senescence [18] and disuse [16]. In addition, IGFBP-5 overproduction induces retarded growth and reduced skeletal muscle mass [48]. Furthermore, when IGFBP-5 was directly applied to paralyzed muscles, it prevented cell proliferation stimulation [36]. Conversely, studies have also shown that IGFBP-5 can promote cell survival and protect against apoptosis in myoblasts [49,50]. In our study, IGFBP-5 expression increased from day 1 after denervation, and it was maintained on successive days. To our knowledge, only one study reported the effect of denervation on IGFBPs, but only on day 2 after denervation, and IGFBP-5 levels were also increased [19]. Here, we further found that this increase was maintained at least 14 days after denervation. IGFBP-5 is able to suppress the biological activities of IGF-1, both in vitro and in vivo [36]. Therefore, IGFBP-5 could have an important role in muscle atrophy by opposing the IGF-1 effects. Furthermore, when muscle atrophy is prevented, either with an anti-inflammatory drug [51], the intake of a nutraceutical [18] or by muscle overload [16], IGFBP-5 levels were normalized, reinforcing the role of this protein in muscle atrophy.

The atrophy-associated IGFBP-5 overexpression may be induced by several factors. In muscle atrophy induced by inflammation, the inflammatory mediators (such as cytokines and prostaglandins) could mediate this stimulation [17]. However, in the present study, the stimulatory effect on IGFBP-5 may be related to the reduction in muscle loa-ding. In this sense, Awede et al. [16] demonstrated that the expression of IGFBP-5 in the soleus muscle is dependent on muscle loading, with an increase observed under loading conditions, while it decreased under overload. Thus, mechanical transduction could be another regulatory mechanism controlling IGFBP-5 expression.

The different role of IGFBP-5 may be determined by the PAPP-A expression [14], as this protease, by cleaving IGFBP-5, might facilitate the release and therefore the action of IGF-1. In the present study, denervation induced an increase in PAPP-A and PAPP-A2 within the 24 h after surgery. To our knowledge, this is the first time that this response has been described in denervated muscle. Other authors have reported a 10-fold increase of PAPP-A three and five days after muscle crush injury [26], and they related this increase to the process of muscle regeneration. Interestingly, these authors also found decreased expression of these enzymes at day 14 after muscle damage. The stimulatory effect of denervation on PAPP-A is transient; after an initial peak, its levels decreased at 7–14 days after denervation. As mentioned above, PAPP increase could lead to an increase in the local concentration of free IGF-1, which in turn would promote proliferation and subsequent fusion of myoblasts/satellite cells into regenerating myofibers. In this sense, proliferation of myoblasts in vitro has been reported after treatment with PAPP-A [25]. In fact, it has been proposed that IGFBP proteolysis can lead to greater anabolic responses than local IGF-1 production [26]. The decreased levels of these proteases on days 3–7 after denervation may be the consequence of the damage being irreparable. The factors involved in this PAPPs regulation are unknown and deserve further investigation. Interestingly, the ratio IGFBP-5/PAPP-A increased significantly at day 3 after denervation, and this increase was maintained during the following days. This pattern of expression was strongly correlated with atrophic parameters such as myogenin (both mRNA and protein), MuRF-1 mRNA and LC3b-II protein. The IGFBP-5 knockdown myoblasts presented a marked reduction in myogenin [52], demonstrating the important regulation of IGFBP-5 on myogenin expression.

Our results show that denervation did not affect the gene expression of stanniocalcins 1 y 2 (STC1 and 2) in the gastrocnemius muscle, which seems to indicate that they are not involved in the changes of the expression of PAPPs and the IGF-1 system. Several studies indicated the possible involvement of these enzymes in regulating growth and IGF-1 activity (for review [53]). However, there are also contradictory data about GH/IGF-1 axis involvement in transgenic STCs mouse models. It should be noted that neither the STC2 transgenic mice [54] nor STC2 knockout mice [55] show any alteration in circulating IGF-1. Even though Jepsen et al. [29] described changes in cleavage of IGFBP-4 in a culture of embryonic fibroblasts derived from STC2 transgenic mice. Therefore, despite several reports indicating that STCs play a role in growth, its contribution by modulating the bio-disponibility of local IGF-1 during muscle atrophy remains to be established.

This study has some limitations. Mainly, it lacks mechanistic analysis; it would be interesting to measure the enzymatic activity of PAPP-A to ensure its role in the first days after denervation. Further studies using inhibitors of this enzyme could reveal the weight of this enzyme in this process.

In summary, our study showed that muscle IGF-1 increased shortly after denervation; this fact, together with the increase in IGFBP-5 proteases (PAPP-A and PAPP-A2), may facilitate IGF-1 action on muscle growth and regeneration on the first few days after the damage. Subsequently, decreased expression of PAPPs and continued overexpression of IGFBP-5 could facilitate the proteolytic and muscle wasting mechanisms that end up predominating 7 days after denervation.

## 4. Materials and Methods

### 4.1. Animals and Experimental Protocol

Studies were performed on male Wistar rats (Charles River, Wilmington, MA, USA) weighing 200–250 g. They were housed under a temperature- and humidity-controlled environment with a 12 h light–dark cycle and free access to food and water. All experimental procedures were approved by the Animal Care and Use Committee of Universidad Complutense de Madrid (protocol code PROEX 038.0/21), following the guidelines of the European Union Council (2010/63/EU) and the Spanish Government (118/2021) for the use of animals in research.

Sciatic nerve denervation was performed under isoflurane anesthesia. A 2 cm skin incision was made in the mid-posterolateral area of the thigh. A 10–12 mm pocket was made along the separation of the muscle groups or bundles and deepened by blunt dissection to expose the sciatic nerve. The sciatic nerve of the right hind limb was then cut. For the sham operation, the sciatic nerve of the left hind limb was exposed but not transected. The muscle groups were approximated and returned to close the created pocket, and the skin incision was closed with surgical staples. After denervation, a lack of active locomotion was observed in the right paw, from which the nerve was cut off. Thirty-nine animals underwent sciatic nerve of the right hind limb, whereas the left hind limb was sham operated. Rats were euthanized and sacrificed 1, 3, 7 and 14 days after denervation. Another group of control rats was submitted to sham surgery in both hind limbs and was euthanized and sacrificed 7 days after surgery. All animals were daily weighed. Both gastrocnemius muscles were removed, dissected and rapidly frozen in liquid nitrogen and stored at −80 °C until analysis.

### 4.2. Skeletal Muscle Histology

The medial part of the gastrocnemius muscles was placed on a transparent film, glued at one end to a cork with gum tragacanth (Fibraguar; Fardi, Madrid, Spain). Samples were frozen in cooled isopentane using liquid nitrogen and stored at −80 °C. Cryostat sections of 10 μm were fixed with 4% PFA and stained with hematoxylin-eosin. Digital images were acquired with a Leica DMI300 microscope and Image J 1.8.0 software was used to measure fiber area.

### 4.3. Quantitative Real-Time Polymerase Chain Reaction (RT-qPCR)

The Trisure (BIOLINE, London, UK) protocol was used to extract all RNA from the gastrocnemius. Isolated RNA was quantified using a BioPhotometer spectrophotometer (Eppendorf International, Hamburg, Germany) and its integrity was confirmed using agarose gel electrophoresis stained with GelRed (Biotium, Hayward, CA, USA). One microgram of total RNA was retrotranscribed into cDNA using a High-Capacity cDNA Reverse Transcription Kit (Applied Biosystems, Thermo Fisher Scientific, Madrid, Spain). The resulting cDNA samples were used with specific primers (Table 1) and 1 × Takara SYBR Green Premix Ex Taq (Takara BIO Inc., Otsu, Japan) to perform real time-PCR. The PCR cycles were 95 °C for 10 min, 95 °C for 15 s and 60N °C for 1 min (total 40 cycles). A melting curve was also performed in order to verify the specificity of the amplification. 18S ribosomal RNA were used as the invariant control, and relative gene expression was determined by usingthe 2^−∆∆Ct^ method.

### 4.4. Western Blot

For protein analysis in the gastrocnemius, 10 µL/mg of RIPA lysis buffer (containing a protease inhibitor mix: phenylmethane sulfonyl fluoride 100 mM, sodium deoxycholate 12.5 mM, sodium orthovanadate 12.5 mM, and with phosphatase inhibitors, all from Sigma-Aldrich, St. Louis, MO, USA) was added to 100 mg of muscle. Lysates were incubated for 20 min at 4 °C and centrifugated at 13,000 r.p.m. for 20 min. The Bradford protein quantification assay (Sigma-Aldrich, St. Louis, MO, USA) was used to determine the protein concentration. A mix of protein extracted and Laemmli loading buffer (Bio-Rad, Madrid, Spain) (1:1) was boiled at 95 °C for 5 min. 50 µg of protein was subjected to electrophoresis using polyacrylamide 4–20% gradient gels (Bio-Rad, Madrid, Spain) under reducing conditions at 100–200 V for 90 min. Separated proteins were transferred onto a nitrocellulose membrane and then blocked using 5% non-fat dry milk and 0.1% Tween (Sigma-Aldrich) in Tris-buffered saline (TBS). In order to analyze the transfer efficiency, membranes were stained with Ponceau-S (Bio-Rad, Madrid, Spain). After blocking the membranes with TBS containing 5% (*w*/*v*) non-fat dried milk, they were incubated overnight at 4 °C with primary antibodies: HDAC4 (antibody ID: 7628, 1:2000; Cell Signaling Technology; Danvers, MA, USA); Myogenin (antibody ID: sc-12732, 1:500; Santa Cruz Biotechnology; Dallas, TX, USA); LC3b (antibody ID: 12741, 1:1000; Cell Signaling Technology; Danvers, MA, USA) and alpha 2 spectrin (antibody ID: sc-48382; 1:1000; Santa Cruz Biotechnology, Dallas, TX, USA). Secondary antibodies conjugated with horseradish peroxidase used to detect primary antibodies were anti-mouse immuno-globulin G (IgG) (Amersham Biosciences; Little Chalfont, UK), and anti-rabbit IgG (GE Healthcare; Chicago, IL, USA). Peroxidase activity (enhanced chemiluminescent reagent from Amersham Biosciences, Little Chalfont, UK) was visualized with densitometry using Gene Tools Analysis 4.3.14 software.

### 4.5. Statistical Analysis

A one-way ANOVA followed by the post hoc LSD test was performed to compare groups at different days. Single comparisons between the right and left muscles were assessed with Student’s *t* test. Simple correlations were assessed using Pearson’s correlation coefficients. Data were presented as the mean ± standard error of the mean (SEM). All analyses were performed using SPSS 25 for Windows. A *p*-value of <0.05 was considered statistically significant.

## 5. Conclusions

Denervation-induced atrophy is a process in which the IGF-1/IGFBP-5/PAPP system seems to play a fundamental role. The increase in both IGF-1 and PAPP-A expression could facilitate the regenerative process during the first days after denervation. However, the subsequent decrease of PAPP-A expression and the increased levels of IGFBP-5, could intensify the process of muscle wasting.

## Figures and Tables

**Figure 1 ijms-24-14112-f001:**
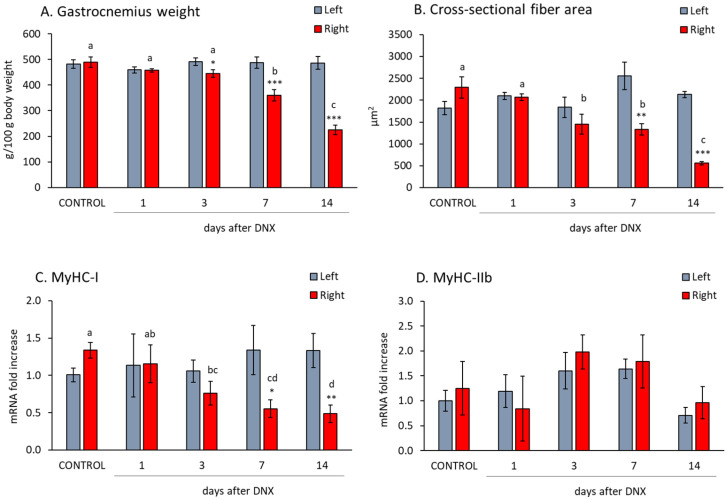
The effect of sciatic denervation on relative gastrocnemius weight (**A**), cross-sectional fiber area (**B**), and mRNA levels of myosin heavy chain isoforms (MHC) I and IIb (**C**,**D**), at different time points post-denervation (DNX). Values are represented as mean ± standard error of the mean (SEM) of left (SHAM operated) and right (denervated) paws in comparison with control animals (SHAM operated in both hind limbs). Statistics: *** *p* < 0.001, ** *p* < 0.01 and * *p* < 0.05 in the comparisons between right vs. left paw with Student’s *t* test; a ≠ b ≠ c ≠ d in the comparisons between different time-points groups of the right paw by least post hoc analysis after significant one-way ANOVA.

**Figure 2 ijms-24-14112-f002:**
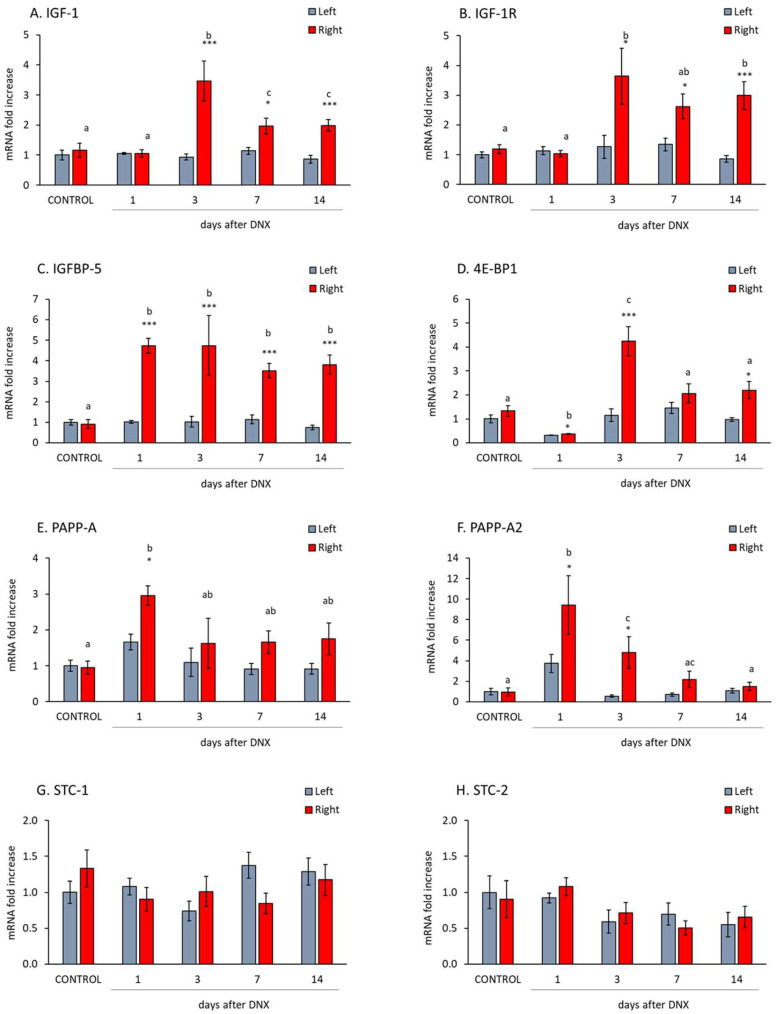
Effect of sciatic denervation on gastrocnemius mRNA levels of insulin-like growth factor 1 (IGF-1) (**A**), IGF-1 receptor (IGF-1R) (**B**), IGF-1 binding protein 5 (IGFBP-5) (**C**), eukaryotic initiation factor 4E-binding protein 1 (4E-BP1) (**D**), pregnancy-associated plasma protein-A (PAPP-A) (**E**) and PAPP-A2 (**F**), stanniocalcin 1 (STC-1) (**G**) and STC-2 (**H**) at different days after denervation (DNX). Values are represented as mean ± standard error of the mean (SEM) of left (SHAM operated) and right (denervated) paws in comparison with control animals (SHAM operated in both hind limbs). Statistics: *** *p* < 0.001 and * *p* < 0.05 in the comparisons between right vs. left paw with Student’s *t* test; a ≠ b ≠ c in the comparisons between different time-points groups of the right paw with least post hoc analysis after significant one-way ANOVA.

**Figure 3 ijms-24-14112-f003:**
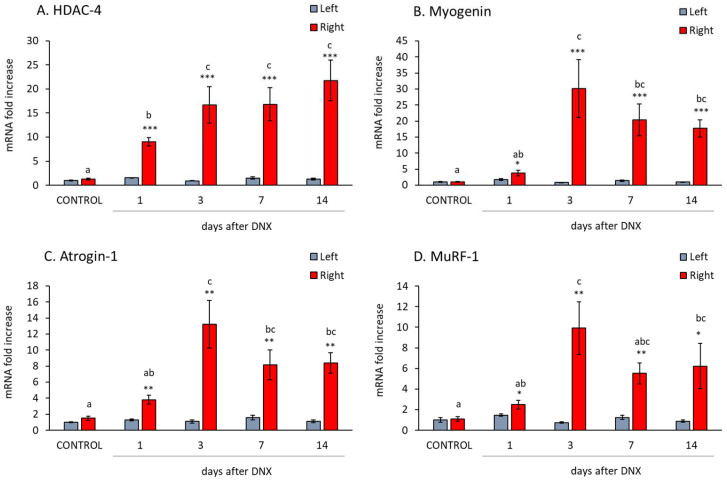
Effect of sciatic denervation on gastrocnemius mRNA levels of histone deacetylase 4 (HDAC-4) (**A**), myogenin (**B**), atrogin 1 (**C**), muscle RING-finger protein-1 (MuRF-1) (**D**) at different time points post-denervation (DNX). Values are represented as mean ± standard error of the mean (SEM) of left (SHAM operated) and right (denervated) paws in comparison with control animals (SHAM operated in both hind limbs). Statistics: *** *p* < 0.001, ** *p* < 0.01 and * *p* < 0.05 in the comparisons between right vs. left paw with Student’s *t* test; a ≠ b ≠ c in the comparisons between different time-points groups of the right paw with least post hoc analysis after significant one-way ANOVA.

**Figure 4 ijms-24-14112-f004:**
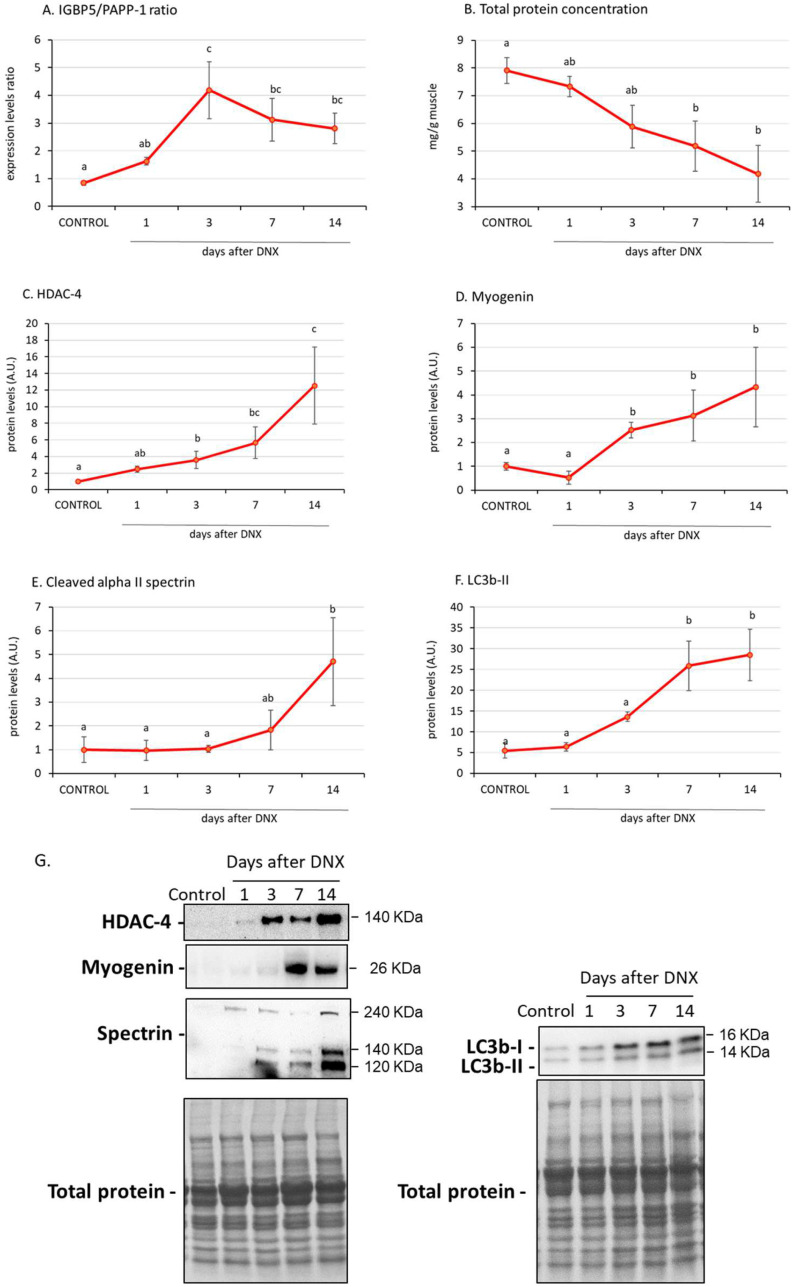
Effect of sciatic denervation on the IGFBP-5/PAPP-A expression levels ratio (**A**), total protein concentration (**B**) and the protein levels of HDAC-4 (**C**), myogenin (**D**), cleaved alpha II spectrin (**E**) and light chain 3b II (LC3b-II) (**F**) at different time points post-denervation (DNX). Values are represented as mean ± standard error of the mean (SEM) of right (denervated) paws in comparison with control animals (SHAM operated in both hind limbs). Representative Western blots are shown in (**G**). Statistics: a ≠ b ≠ c in the comparisons between different time-point groups of the right paw by least post hoc analysis after significant one-way ANOVA.

**Figure 5 ijms-24-14112-f005:**
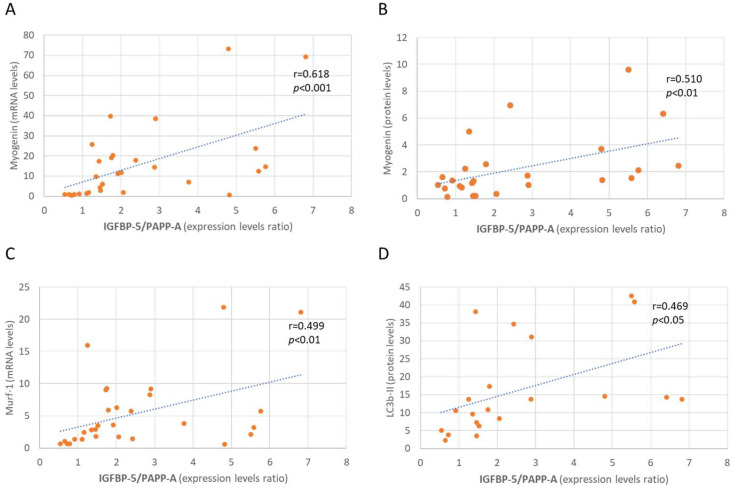
Linear correlation between IGFBP-5/PAPP-A expression levels ratio and myogenin mRNA (**A**) and protein levels (**B**), MuRF-1 expression levels (**C**) and LC3b-II protein levels (**D**). Statistics: Pearson’s correlations.

**Table 1 ijms-24-14112-t001:** Primers for real-time PCR.

Gene	Forward Primer (5′ to 3′)	Reverse Primer (5′ to 3′)
Atrogin-1	GAACAGCAAAACCAAAACTCAGTA	GCTCCTTAGTACTCCCTTTGTGAA
HDAC4	CACACCTCTTGGAGGGTACAA	AGCCCATCAGCTGTTTTGTC
IGF-1	GCTATGGCTCCAGCATTCG	GGATGAGTGTTGCTTCCGGA
IGF-1R	GCCTCCAACTTTGTCTTTGC	TCACTGGGCCAGGAATGT
IGFBP5	GGCGAGCAAACCAAGATAGA	GGTCTCCTCAGCCATCTCG
MuRF-1	TGTCTGGAGGTCGTTTCCG	AAGTGATCATGGACCGGCAT
MyHC-I	CCAAGGGCCTGAATGAAGAGT	TGTTTCTGCCTAAGGTGCTGT
MyHC-IIb	TCCTATTTTCTGGGGACAA	ACCCTTCTTCTTGCCACCTT
Myogenin	CCTTGCTCAGCTCCCTCA	TGGGAGTTGCATTCACTGG
PAPP-A	CTCCACACAGAGCCTACTTGG	ATTAGGGCCTCCTTGTCCCA
PAPP-A2	CCCTCCCCCATCTGTACTCA	TGGAGTAGATGAGCCCGGAA
STC-1	CAACAGTGCCCTACAGGTT	TCCCATTGGCGATGCACTTT
STC-2	CACCGACCACCACCTAACAG	CTCACTGCTTCCAGAGGGTC
4E-BP1	ACTAGCCCTACCAGCGATGA	AGCATCACTGCGTCCTATGG
18S	GGTGCATGGCCGTTCTTA	TCGTTCGTTATCGGAATTAACC

## Data Availability

Not applicable.

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
