# Peer review of "Time-Dependent Changes in Muscle IGF1-IGFBP5-PAPP System after Sciatic Denervation"

_ijms, 2023, doi:10.3390/ijms241814112_

Round 1
Reviewer 1 Report
Manuscript title:
Time dependent changes in muscle IGF1-IGFBP5-PAPP system after sciatic denervation
Ana Isabel Martín, Álvaro Moreno-Ruperez, Elena Nebot, Miriam Granado, Daniel Jaque, María Paz Nieto Bona, Asunción López-Calderón and Teresa Priego
The manuscript is good written and the subject is novel. Please answer the following comments:
1. Line 30; correct the ”papppaylsins”
2. Lines 62-63: The sentence structure needs improvement.
3. Line 78: The “anti-anabolic and pro-catabolic effects of IGFBP-5” should be more described.
4. Line 102: Rewrite the “There is one extra family of regulators within the IGF system: stanniocalcins.”
5. A table summarizing a number of update references in sciatic denervation can improve the paper.
6. The conclusion is not included in this paper; if it is not the format of journal then it should be added.
Minor editing of English language required
Author Response
Time dependent changes in muscle IGF1-IGFBP5-PAPP system after sciatic denervation
Ana Isabel Martín, Álvaro Moreno-Ruperez, Elena Nebot, Miriam Granado, Daniel Jaque, María Paz Nieto Bona, Asunción López-Calderón and Teresa Priego
The manuscript is good written and the subject is novel. Please answer the following comments:
1. Line 30; correct the ”papppaylsins”
We have corrected the misspelling, thank you for the remark.
2. Lines 62-63: The sentence structure needs improvement.
We have changed the sentence. Now in lines 53-55:
“The IGF-1 half-life (10-12 minutes) can be prolonged to 12-16 hours, by the binding of this factor to a family of binding proteins called IGF-binding proteins (IGFBPs). In addition, these proteins also regulate IGF-1 bioavailability.”
3. Line 78: The “anti-anabolic and pro-catabolic effects of IGFBP-5” should be more described.
We have decided to remove this sentence to shorten the introduction and make it easy to read. We have also summarized the introduction and the discussion in several parts, as recommended by reviewer 3.
4. Line 102: Rewrite the “There is one extra family of regulators within the IGF system: stanniocalcins.”
The sentence has been rewritten, now in line 87:
“Pappalysins are inhibited by other proteins, the stanniocalcins.”
5. A table summarizing a number of update references in sciatic denervation can improve the paper.
We thank the reviewer for this comment. Although we also think that a summary table can facilitate the understanding of the different studies, we believe that this table is more appropriate in a review article than in an original research article. We have shortened both the introduction and the discussion sections trying to be clearer in our expressions to make it easier to read.
6. The conclusion is not included in this paper; if it is not the format of journal then it should be added.
We have added a conclusion at the end of the article (see lines 413-417), we thank the reviewer for the suggestion.
“Denervation-induced atrophy is a process in which the IGF-1/IGFBP-5/PAPP system seems to play a fundamental role. The increase in both IGF-1 and PAPP-A expression could facilitate the regenerative process during the first days after denervation. However, the subsequent decrease of PAPP-A expression and the increased levels of IGFBP-5, could intensify the process of muscle wasting.”
Reviewer 2 Report
The authors focused on investigating the time-dependent changes in the IGF1-IGFBP5-PAPP system and its regulators in gastrocnemius muscle after sciatic denervation. Interesting topic and good results part offering the perspective of a promising paper. Overall, this paper is written in a concise and orderly manner with sufficient introduction, detailed methods and solid data. The article is easy to read, well designated and presented, and can be of interest to reader and researchers. Very well-chosen statistical analysis methods. However, I have the following suggestions related to the improvements that should be added:
- There are spelling, punctuation and some grammar issues (e.g: lines: 145-148, 385, etc). This will apply to the whole manuscript.
- Please summarize the main theme of the article in a graphical abstract.
Minor editing of English language required.
Author Response
The authors focused on investigating the time-dependent changes in the IGF1-IGFBP5-PAPP system and its regulators in gastrocnemius muscle after sciatic denervation. Interesting topic and good results part offering the perspective of a promising paper. Overall, this paper is written in a concise and orderly manner with sufficient introduction, detailed methods and solid data. The article is easy to read, well designated and presented, and can be of interest to reader and researchers. Very well-chosen statistical analysis methods. However, I have the following suggestions related to the improvements that should be added:
- There are spelling, punctuation and some grammar issues (e.g: lines: 145-148, 385, etc). This will apply to the whole manuscript.
We thank the reviewer for the comment, we have fully revised the entire manuscript and corrected the spelling, punctuation, and grammar errors.
- Please summarize the main theme of the article in a graphical abstract.
We submitted the graphical abstract together with the images of the article in the first version. However, we have added the mentioned graphical abstract at the end of the main document (see line 582) to be easily checked by the reviewer.
Reviewer 3 Report
The manuscript of Martín et al. presents changes in mRNA expression of the family of proteins involved in the IGF-1 system: including IGF-1 itself, IGFBP-5, EIF4BP1PAPP-A and PAPP-A2, PAPP inhibitors, STC-1 and 2, HDAC-4 myogenins, myogenin, atrogin-1 and MuRF-1. Sciatic denervation in mice up to 14 days after denervation has been used as a model. Also, western Blot has been performed for some selected proteins.
There are some interesting aspectsof this study, however, I would suggest the following changes:
1) It seems that some parts of the study are a repetition of the previous studies. I would neither describe these parts in detail nor put them in the figures, especially if not significant.
E.g. some of the RNA studies and patomorphology have already been performed (Discussion line 275, Results line 117)
2) Please specify which mRNA studies have been performed in which timepoint by other authors (a table would do well in this case) and which are new.
3) Are there any similar studies using proteomics and RNA-Seq?
4) I would try to shorten the discussion and maybe also the introduction part. Some information could be put in a table or on a figure.
5) Are there any corresponding results from other models of muscle atrophy?
6) Please list the limitations of the study, especially the limitations of the methods applied.
Author Response
The manuscript of Martín et al. presents changes in mRNA expression of the family of proteins involved in the IGF-1 system: including IGF-1 itself, IGFBP-5, EIF4BP1PAPP-A and PAPP-A2, PAPP inhibitors, STC-1 and 2, HDAC-4 myogenins, myogenin, atrogin-1 and MuRF-1. Sciatic denervation in mice up to 14 days after denervation has been used as a model. Also, western Blot has been performed for some selected proteins.
There are some interesting aspects of this study, however, I would suggest the following changes:
1) It seems that some parts of the study are a repetition of the previous studies. I would neither describe these parts in detail nor put them in the figures, especially if not significant.
E.g. some of the RNA studies and patomorphology have already been performed (Discussion line 275, Results line 117)
Although some results of denervation, in terms of muscle weight and fiber area, have been previously described by several authors, we think that it is important to show these results as verification that the study design has been correctly carried out. Moreover, these results are considered outstanding in order to know how is the evolution of the atrophic process, and also to compare and correlate the atrophy degree with modifications of the new factors.
2) Please specify which mRNA studies have been performed in which time point by other authors (a table would do well in this case) and which are new.
In the discussion of the revised manuscript (pg. 9, line 268) we pointed out that “To our knowledge, only one study reported the effect of denervation on IGFBPs, but only on day 2 after denervation, and IGFBP-5 levels were also increased. In our study, we further found that this increase was maintained at least 14 days after denervation”. In the case of pappalysins and stanniocalcins, to the best of our knowledge, no studies have been performed. Thus, all the mRNA studies on these proteins could be considered new findings.
3) Are there any similar studies using proteomics and RNA-Seq?
Yes, there are similar studies where authors have used omics techniques to explore the changes induced by denervation. For example: Shen et al. 2019 (doi: 10.3389/fphys.2019.01298), performed a microarray study in which described the importance of the inflammatory process during the atrophy. They also showed an increase in the expression of positive regulation of growth (within the first hours after denervation) followed by a decrease of these genes afterwards. There is also a study conducted by Sun et al. (2014) (https://doi.org/10.3892/ijmm.2014.1737) in which a proteomic and bioinformatic analysis were performed in tiabilis anterior muscle at 1 and 4 weeks after denervation. They described important changes in the protein pattern. Specifically, they highlighted the involvement of a protein, the tumor necrosis factor receptor-associated factor-6 (TRAF6), as an important factor involved in the atrophy. We have mentioned these studies in the manuscript (pg 1). However, these authors did not perform any analysis on the IGF1-BP-PAPP family of proteins.
4) I would try to shorten the discussion and maybe also the introduction part. Some information could be put in a table or on a figure.
Both introduction and discussion have been shortened in order to clarify all the information.
5) Are there any corresponding results from other models of muscle atrophy?
An increase of IGF-I in muscle has been reported in several model of muscle atrophy (as described in the paper, lines 219-236). However, a limited number of studies evaluated the expression of IGFBP-5 in skeletal muscle. As mentioned in the introduction (pg. 2 of the revised manuscript), increased expression of IGFBP-5 in muscle has been also reported in muscle atrophy induced by disuse, aging and experimental arthritis. Nevertheless, to our knowledge, there are no data on the relationship between muscle atrophy and pappalysins or stanniocalcins expression in any model of muscle atrophy.
6) Please list the limitations of the study, especially the limitations of the methods applied.
Limitations of the study have been included in pg 10, lines 320-323.
Round 2
Reviewer 3 Report
Thank you for your answers, I am satisfied with the responses and changes introduced